# Activation of Discs large by aPKC aligns the mitotic spindle to the polarity axis during asymmetric cell division

Ognjen Golub[†], Brett Wee[†], Rhonda A Newman, Nicole M Paterson, Kenneth E Prehoda*

Institute of Molecular Biology, Department of Chemistry and Biochemistry, University of Oregon, Eugene, United States

**Abstract** Asymmetric division generates cellular diversity by producing daughter cells with different fates. In animals, the mitotic spindle aligns with Par complex polarized fate determinants, ensuring that fate determinant cortical domains are bisected by the cleavage furrow. Here, we investigate the mechanisms that couple spindle orientation to polarity during asymmetric cell division of *Drosophila* neuroblasts. We find that the tumor suppressor Discs large (Dlg) links the Par complex component atypical Protein Kinase C (aPKC) to the essential spindle orientation factor GukHolder (GukH). Dlg is autoinhibited by an intramolecular interaction between its SH3 and GK domains, preventing Dlg interaction with GukH at cortical sites lacking aPKC. When co-localized with aPKC, Dlg is phosphorylated in its SH3 domain which disrupts autoinhibition and allows GukH recruitment by the GK domain. Our work establishes a molecular connection between the polarity and spindle orientation machineries during asymmetric cell division.

DOI: https://doi.org/10.7554/eLife.32137.001

*For correspondence:
prehoda@uoregon.edu

[†]These authors contributed equally to this work

Competing interests: The authors declare that no competing interests exist.

## Introduction

Animals are unique in their degree of multicellular organization, with diverse cell types structured into complex tissues and organs. The cellular mechanisms that generate diversity are critical for the development of these highly organized structures and their maintenance during homeostasis. Asymmetric division is a common mechanism for generating cellular diversity, as it produces daughter cells that assume distinct fates (*Knoblich, 2010*; *Roubinet and Cabernard, 2014*). A critical aspect of the molecular mechanism underlying this process is the segregation of unique fate determinants into the resulting daughter cells. As the asymmetrically dividing cell proceeds through mitosis, unique sets of fate determinants become polarized into discrete cortical domains that are bisected by the cleavage furrow as mitosis is completed (*Gönczy, 2008*; *Nance and Zallen, 2011*). Thus, a key step in asymmetric cell division is the coordination of the polarity axis with the processes that position the cleavage furrow. Because furrow position is controlled by the mitotic spindle (*Barr and Gruneberg, 2007*; *Fededa and Gerlich, 2012*; *Glotzer, 2005*), segregation of fate determinants during asymmetric cell division requires precise coupling between cell polarity and spindle orientation machinery. Here, we use an induced polarity cultured cell system and *Drosophila* neuroblasts to uncover a mechanism for linking polarity and spindle position during asymmetric cell division.

Neuroblasts populate the fly central nervous system by undergoing repeated asymmetric divisions during embryonic and larval developmental stages (*Gallaud et al., 2017*; *Knoblich, 2010*). At the completion of a typical division, one daughter cell retains the neuroblast fate (i.e. self-renewal), whereas the other assumes a differentiated fate (e.g. neuron). The molecular components that specify distinct daughter cell fates form domains opposite one another on the cell cortex. The basal cortical domain contains molecules important for specifying neuronal fate, such as Miranda, Brat, and

Prospero. The apical cortical domain contains a number of regulatory proteins including the Par polarity complex, which restricts the neuronal fate determinants to the basal domain (*Atwood and Prehoda, 2009*; *Bailey and Prehoda, 2015*; *Wirtz-Peitz et al., 2008*). This domain also contains proteins that align the spindle along the apical-basal polarity axis, such as Partner of Inscuteable (Pins) and the tumor suppressor Discs large (Dlg) (*Lu and Johnston, 2013a*; *Roubinet and Cabernard, 2014*). However, Dlg is also found at non-apical cortical regions (*Albertson and Doe, 2003*) suggesting that other mechanisms besides polarization are likely to be necessary to ensure its activity is restricted to the apical cortex. Here, we investigate how polarity is coupled to Dlg's spindle orientation activity.

Dlg is a member of the Membrane Associated Guanylate Kinase (MAGUK) family of proteins that regulate diverse cellular processes including adhesion and neuronal synapse formation (*Anderson et al., 2016*; *Oliva et al., 2012*). Like other MAGUKs, Dlg contains a GK protein interaction module that binds downstream effector proteins, such as the kinesin Khc73 (*Figure 1A*) (*Albertson and Doe, 2003*; *Lu and Prehoda, 2013b*; *Newman and Prehoda, 2009*). The Dlg GK domain is required for neuroblast spindle orientation (*Siegrist and Doe, 2005*), presumably because of its role in recruiting these effectors. Binding of certain GK targets can be blocked, however, by an autoinhibitory intramolecular interaction between the GK and an adjacent SH3 domain (*Johnston et al., 2009*; *Marcette et al., 2009*; *McGee et al., 2001*). Analysis of Dlg function in spindle orientation suggests that autoinhibition plays a critical, albeit paradoxical, role in the process. In cultured *Drosophila* S2 cells, polarized Dlg GK induces spindle alignment, but polarized SH3GK does not (*Marcette et al., 2009*), suggesting that the intramolecular interaction inhibits Dlg's spindle orientation activity. However, the intramolecular interaction is required for Dlg function in vivo as neuroblasts containing a *dlg* allele that lacks the interaction (*dlg^sw*) is non-functional (*Newman and Prehoda, 2009*). Thus, autoinhibition prevents Dlg's spindle orientation activity but is required for spindle positioning in vivo, suggesting that regulation of Dlg's GK by its SH3 domain is critically important for Dlg function. How this regulatory interaction might be modulated to activate Dlg is unknown, as are the Dlg spindle orientation effectors regulated by the interaction (*Figure 1A*). As coupling of polarity and spindle orientation is a key aspect of asymmetric cell division, we sought to

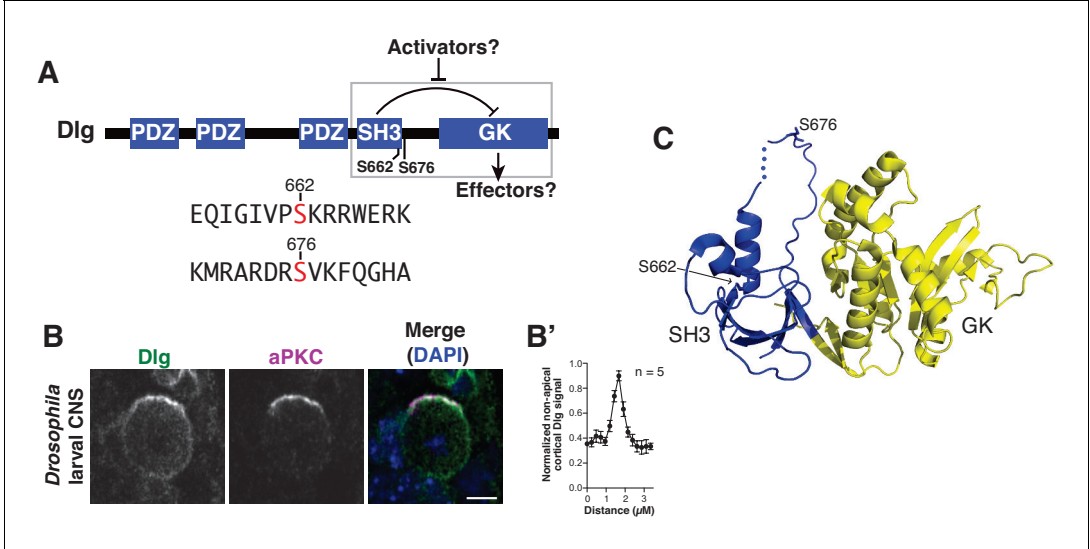

**Figure 1.** Discs large localizes uniformly to the neuroblast cortex and is phosphorylated by atypical Protein Kinase C. (**A**) Dlg domain structure. An intramolecular interaction between the SH3 and GK domains inhibits GK binding to certain effectors. Dlg activators may function by disrupting the intramolecular interaction, allowing a necessary spindle orientation factor to bind. S662 and S676 denote the two sites of modification by aPKC identified by mass spectrometry. The sequences surrounding each phosphorylated serine are shown. (**B**) Localization of Dlg and aPKC in a metaphase *Drosophila* larval brain neuroblast showing that Dlg, while enriched at the apical cortex with aPKC, is also found in significant amounts at non-apical regions of the cortex. (**B'**) Quantification of non-apical Dlg signal. (**C**) Location of aPKC phosphorylation sites mapped on to the structure of the SH3GK module from PSD-95 (PDB ID: 1KJW).
DOI: https://doi.org/10.7554/eLife.32137.002

identify the mechanism by which the Dlg intramolecular interaction is regulated and how its regulation might influence these important cellular processes.

## Results

### aPKC directly phosphorylates the Dlg SH3 and Hook domains

As described above, an intramolecular interaction within Dlg inhibits its spindle orienting activity, yet Dlg autoinhibition is required for spindle orientation in vivo (*Marcette et al., 2009*; *Newman and Prehoda, 2009*). In cells such as *Drosophila* neuroblasts, Dlg is localized at the apical cell cortex where the spindle is aligned, but also elsewhere on the cortex (*Figure 1B,B'*) (*Siegrist and Doe, 2005*), suggesting that it exists in both active (apical) and inactive (non-apical) pools. We reasoned that Dlg may be autoinhibited at non-apical regions but its spindle orienting function activated at the apical cortex. Neuroblasts are polarized by the Par polarity complex whose primary output is the activity of the atypical Protein Kinase C (aPKC) (*Atwood and Prehoda, 2009*; *Knoblich, 2010*; *Prehoda, 2009*). Although aPKC is required for proper spindle alignment (*Guilgur et al., 2012*; *Kim et al., 2009*), the mechanism has been unknown. Canonical PKC has been shown to phosphorylate Dlg during cell migration (*O'Neill et al., 2011*), so we reasoned that aPKC phosphorylation of Dlg might be responsible for activating apical Dlg. To test this hypothesis, we first sought to determine if Dlg is an aPKC substrate and, if so, whether phosphorylation by aPKC might influence Dlg's spindle orientation activity.

We incubated purified Dlg with the kinase domain of aPKC in an in vitro kinase phosphorylation assay. To identify possible modification sites, we subjected the resulting Dlg protein to mass spectrometry analysis following incubation with aPKC. This analysis revealed two primary modified residues, S662 located within the SH3 domain, and S676 located just COOH-terminal to the SH3 (sometimes called the 'HOOK' motif) (*Figure 1A,C*).

### aPKC phosphorylation regulates Dlg SH3GK-mediated spindle orientation activity

To determine if Dlg phosphorylation by aPKC activates its spindle orientation activity, we used the cultured S2-cell-induced polarity spindle orientation assay (*Johnston et al., 2009*). In this assay, expression of the cell adhesion molecule Echinoid (Ed) causes S2 cells to form small clusters with Ed restricted to sites of cell-cell contact. Proteins such as Dlg can be fused to Ed's cytoplasmic domain allowing for their ability to orient the spindle along the induced polarity axis to be examined (*Figure 2A*). Consistent with previous work (*Garcia et al., 2014*; *Johnston et al., 2009*; *Marcette et al., 2009*), we found that S2 spindles are biased toward crescents of Ed fusions with Dlg's GK domain (Ed:GK), whereas spindles in S2 cells expressing Ed:SH3GK were randomly oriented indistinguishable from those expressing Ed:GFP (*Figure 2B–D*).

We next assessed whether Dlg phosphorylation by aPKC can activate spindle orientation by the SH3GK module. While S2 cells expressing Ed:SH3GK alone fail to orient their spindles, coexpression with aPKC induced alignment to SH3GK indistinguishable from that of cells expressing Ed:GK alone (*Figure 2C,E*). Spindle alignment to SH3GK is dependent upon aPKC's catalytic activity, as expression of a kinase dead aPKC^K293W mutant failed to result in spindle alignment to Ed:SH3GK contacts (*Figure 2H*). Furthermore, aPKC expression with the non-phosphorylatable Ed:SH3GK^2A (S662A, S676A) Dlg variant also failed to result in aligned spindles (*Figure 2I*) indicating that modification of these sites is necessary for SH3GK-mediated spindle orientation. Furthermore, phosphorylation at either site is sufficient for this activity as S2 spindles aligned to crescents of Ed:SH3GK variants containing mutations meant to mimic phosphorylation (S662D or S676D) in a manner independent of aPKC (*Figure 2E-G*). Together, these data support a model in which aPKC phosphorylation induces spindle alignment to the SH3GK, presumably by releasing the GK from autoinhibition.

### aPKC regulates Dlg spindle orientation activity in vivo

Loss of aPKC causes the spindle to become misaligned (*Guilgur et al., 2012*; *Kim et al., 2009*), although how aPKC activity is linked to spindle positioning has been unknown. To test whether modification of the aPKC sites within Dlg contributes to spindle orientation in vivo, we assessed the effects of mutations at these sites on neuroblast asymmetric cell division. Previously, we found that

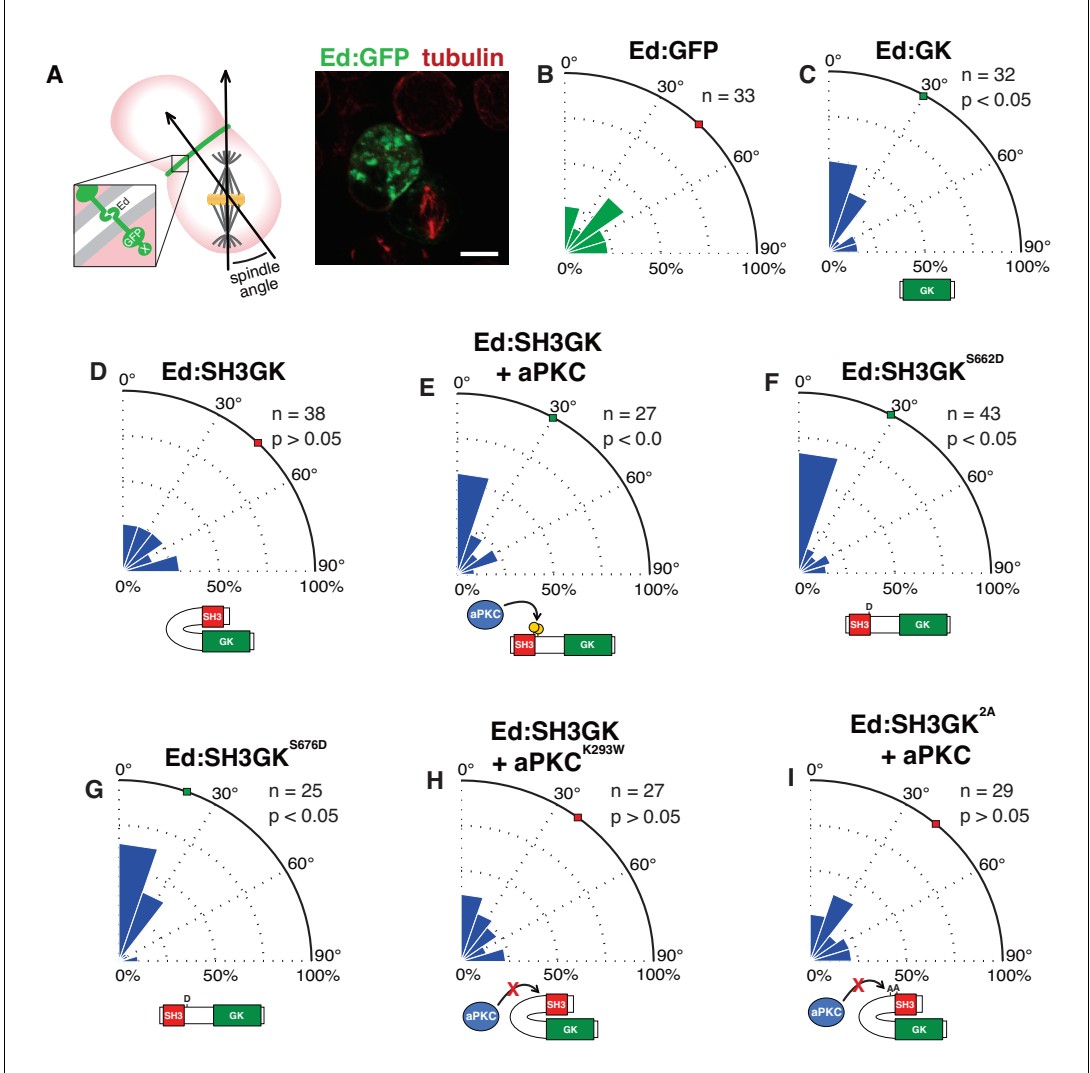

**Figure 2.** The atypical protein kinase C induces spindle orientation by Dlg SH3GK. (**A**) Induced polarity spindle orientation assay. S2 cells clustered by the Echinoid adhesion protein (green) polarize proteins attached to the Ed COOH-terminus ('X'). The spindle angle is measured between axes defined along the spindle and the middle of the polarized crescent and cell. A representative image (anti-tubulin, red; GFP, green) of a typical analyzed cell cluster is shown. Spindle angle measurements (polar histogram; green represents control and blue represents experimental) for cells expressing (**B**) Ed fused to GFP (Ed:GFP) (**C**) Ed fused to the Discs large GK domain (Ed:GK) (**D**) Ed:SH3GK (**E**) Ed:SH3GK co-expressed with aPKC (**F**) Ed:SH3GK with a phosphomimetic aspartic acid at position 662 (**G**) Ed:SH3GK with a phosphomimetic aspartic acid at position 676 (**H**) Ed:SH3GK co-expressed with aPKC containing the K293W mutation in its ATP-binding pocket that prevents kinase activity (**I**) Ed:SH3GK with both aPKC phosphorylation sites mutated to alanine co-expressed with aPKC. For each condition, the number of measurements is shown along with a p-value calculated using a K-S test. Individual measurements are included in comma separated value format in *Figure 2—source data 1*.
DOI: https://doi.org/10.7554/eLife.32137.003

The following source data is available for figure 2:

**Source data 1.** Individual spindle angle measurements in comma separated value format (corresponding figure panel is included in the column header).
DOI: https://doi.org/10.7554/eLife.32137.004

constitutive activation of Dlg using the *dlg^sw* allele disrupts the interaction between the SH3 and GK domains and results in decoupling of spindle orientation from cortical polarity (*Newman and Prehoda, 2009*), indicating that autoinhibition of Dlg is required for its role during mitotic spindle orientation in vivo. Since Dlg is localized both at apical and non-apical cortical regions, we hypothesized that misregulation of Dlg by introducing a phosphomimetic mutation (*Figure 2H*), would result in ectopic Dlg activity and concomitant loss of spindle orientation as observed with *dlg^sw*.

To test this hypothesis, we generated transgenic fly lines harboring Dlg[WT] and Dlg[S662D] transgenes under control of the UAS promoter (the Dlg[2A] variant was toxic, precluding its analysis) and examined their ability to rescue the spindle orientation defect of the *dlg[m52]* null allele. Loss of Dlg results in a moderate spindle orientation phenotype in neuroblasts because of redundancy with the Inscuteable pathway (*Siegrist and Doe, 2005*). We assayed for rescue of the *dlg[m52]* spindle orientation defect in third instar larval neuroblasts (*Figure 3A*), using *worniu*-Gal4-driven expression of UAS-Dlg[WT] and UAS-Dlg[S662D]. We confirm that the *dlg[m52]* allele lacks endogenous Dlg in third instar larval neuroblasts based on immunostaining, which resulted in a reduced precision of spindle alignment with cortical polarity (*Figure 3B,C*) (*Albertson and Doe, 2003*). Exogenous expression of Dlg[WT] fully rescued this defect; however, neuroblasts expressing Dlg containing the phosphomimetic S662D mutation exhibited a spindle orientation defect indistinguishable from *dlg[m52]* mutants (*Figure 3D,E*). This effect is consistent with a requirement for Dlg regulation via modification at this site as the S662D mutation activates Dlg such that its activity is no longer restricted to the apical cortex. We did however observe partial rescue by Dlg[S662D] of the polarity defect that results from the loss of Dlg (*Figure 3E',F*), suggesting that the SH3-mediated regulation of the GK domain is not part of Dlg's role in regulating cell polarity (*Ohshiro et al., 2000*; *Peng et al., 2000*). We conclude that regulation of the SH3GK Dlg module by aPKC is a critical step in coupling apical polarity to spindle orientation in *Drosophila* neuroblasts, as misregulation of this interaction with a constitutively active variant of Dlg results in aberrant spindle positioning in vivo.

## Dlg regulation couples spindle alignment to polarized aPKC

Our data suggest that the Dlg intramolecular SH3GK interaction may function to couple Par complex polarity to spindle orientation, effectively causing the spindle to align to cortical regions containing the Par complex. To directly test this model, we asked whether Dlg could mediate spindle alignment to polarized aPKC crescents in S2 cells. Spindles in S2 cells with polarized Ed:aPKC were randomly oriented (*Figure 4A,B*) indicating that aPKC alone is insufficient to induce alignment. However, coexpression with Dlg SH3GK, which localizes uniformly to the S2 cell cortex (*Figure 4A*), yielded spindle orientation levels similar to Ed:SH3GK activated by soluble aPKC (*Figure 4C*). This observed increase in activity was dependent on aPKC kinase activity, as a kinase dead Ed:aPKC[K293W] expressed with Dlg[SH3GK] did not result in oriented spindles (*Figure 4D*). Furthermore, the non-phosphorylatable Dlg SH3GK[2A] variant also failed to orient the spindle relative to Ed:aPKC crescents (*Figure 4E*). We conclude that the Dlg SH3GK couples cortically polarized aPKC to spindle position via phosphorylation of the SH3 residues, S662 and S676.

## aPKC regulates Dlg's interaction with the GK ligand Gukholder

Our results indicate that aPKC phosphorylation of Dlg's SH3GK module activates Dlg's spindle positioning function. As the SH3 represses the ability of the Dlg GK domain to bind certain ligands, phosphorylation could activate the SH3GK by relieving this inhibition and allowing the GK to bind one or more proteins essential for spindle orientation. We surveyed known GK ligands to determine if their binding is repressed by the SH3 domain (i.e. binds to GK but not SH3GK) and, if so, whether activation by aPKC induces binding to SH3GK. The kinesin Khc73 is a good candidate for an aPKC regulated Dlg ligand because it is essential for Dlg-mediated spindle orientation and binds to Dlg's GK via its MAGUK Binding Stalk (MBS) (*Hanada et al., 2000*; *Lu and Prehoda, 2013b*). However, we compared binding of purified Khc73 MBS to Dlg's GK and SH3GK modules and observed no difference, indicating that its binding is not inhibited by the intramolecular interaction (*Figure 5A*). In contrast, binding of the GK ligand GukHolder (GukH) is strongly repressed by the SH3 domain (*Figure 5A*) (*Marcette et al., 2009*). Modification of either of the aPKC phosphorylation sites with phosphomimetic mutations (S662D or S676D) allows the SH3GK to bind GukH (*Figure 5A*), indicating that regulation of GukH binding could allow for coupling between aPKC phosphorylation and Dlg-mediated spindle orientation. In this model, aPKC induces GukH recruitment to Dlg and GukH is essential for spindle orientation.

We next tested whether aPKC induces GukH recruitment to Dlg. Although GukH is known to localize to the neuroblast apical cortex (*Albertson and Doe, 2003*), the mechanism of its localization has been unknown. We find that while GukH is efficiently recruited to crescents of Ed:GK, it fails to localize to crescents of Ed:GFP or Ed:SH3GK (*Figure 5B–D,J*), suggesting that the Dlg

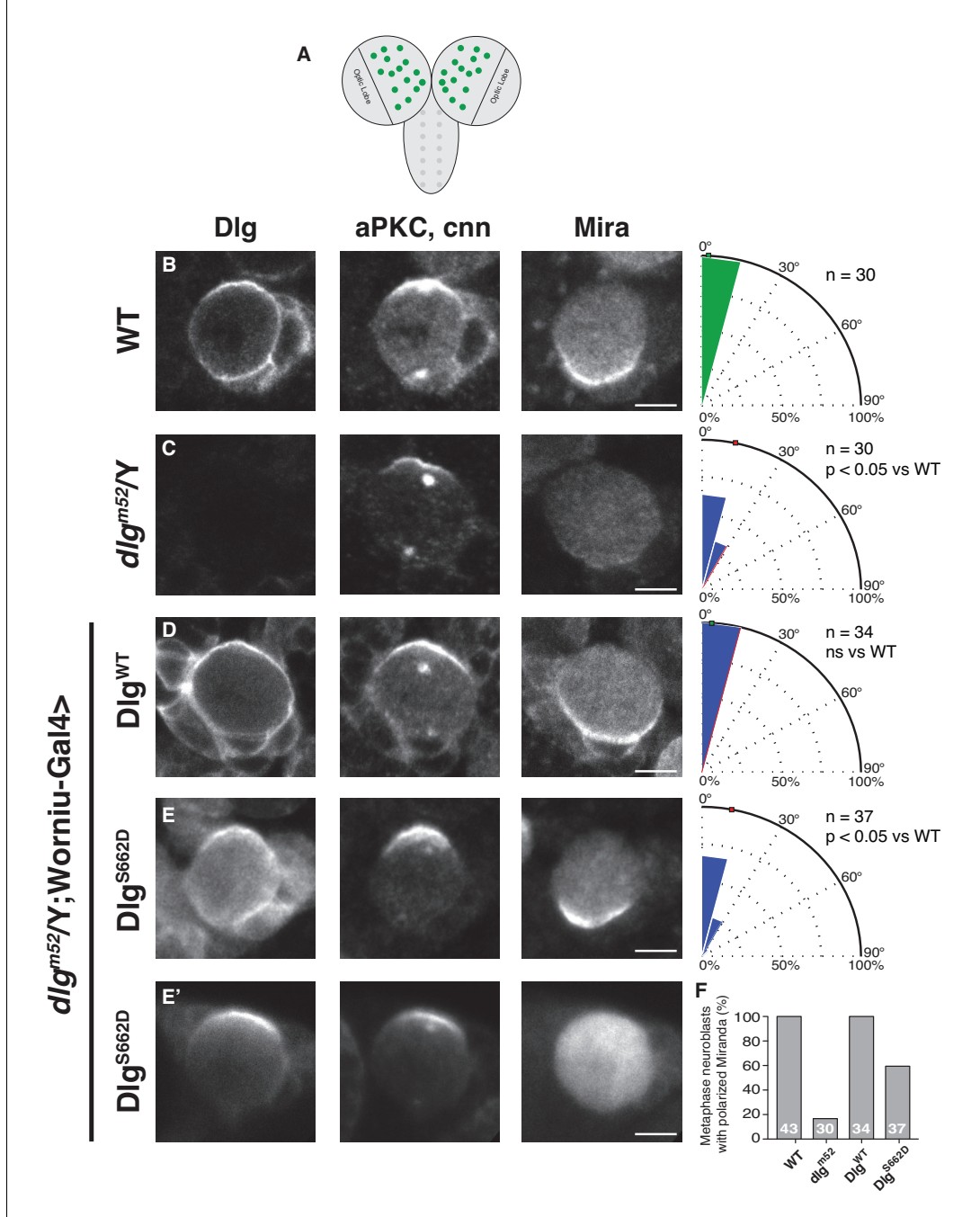

**Figure 3.** Constitutive Dlg activity disrupts spindle orientation in *Drosophila* larval brain neuroblasts. (**A**) Schematic of *Drosophila* larval central nervous system showing brain lobes and ventral nerve cord. In this study we analyzed neuroblasts (green circles) from the brain lobes outside of the optic lobes. Neuroblasts were stained for Dlg, aPKC, the centrosome marker centrosomin (cnn) to measure spindle angle, and the polarized neural fate determinant Miranda (Mira), after dissection from animals with the following genotypes (**B**) Wild-type (**C**) *dlg^{m52}* null mutant (**D**) *dlg^{m52}* Worniu-Gal4 >UAS WT Dlg (**E**) *dlg^{m52}* Worniu-Gal4 >UAS Dlg with a phosphomimetic mutation at site S662 (**E'**) the same genotype as in panel D but showing the variability in Mira polarization we observed. For each condition, a polar histogram with the measured spindle angles is shown along with a p-value calculated by comparison with the wild-type condition using the K-S test. (**F**) Quantification of neuroblasts exhibiting polarized Mira for the conditions shown in panels B-E'. Individual measurements are included in comma separated value format in *Figure 3—source data 1*.

DOI: https://doi.org/10.7554/eLife.32137.005

The following source data is available for figure 3:

**Source data 1.** Individual spindle angle measurements in comma separated value format (corresponding figure panel is included in the column header).
DOI: https://doi.org/10.7554/eLife.32137.006

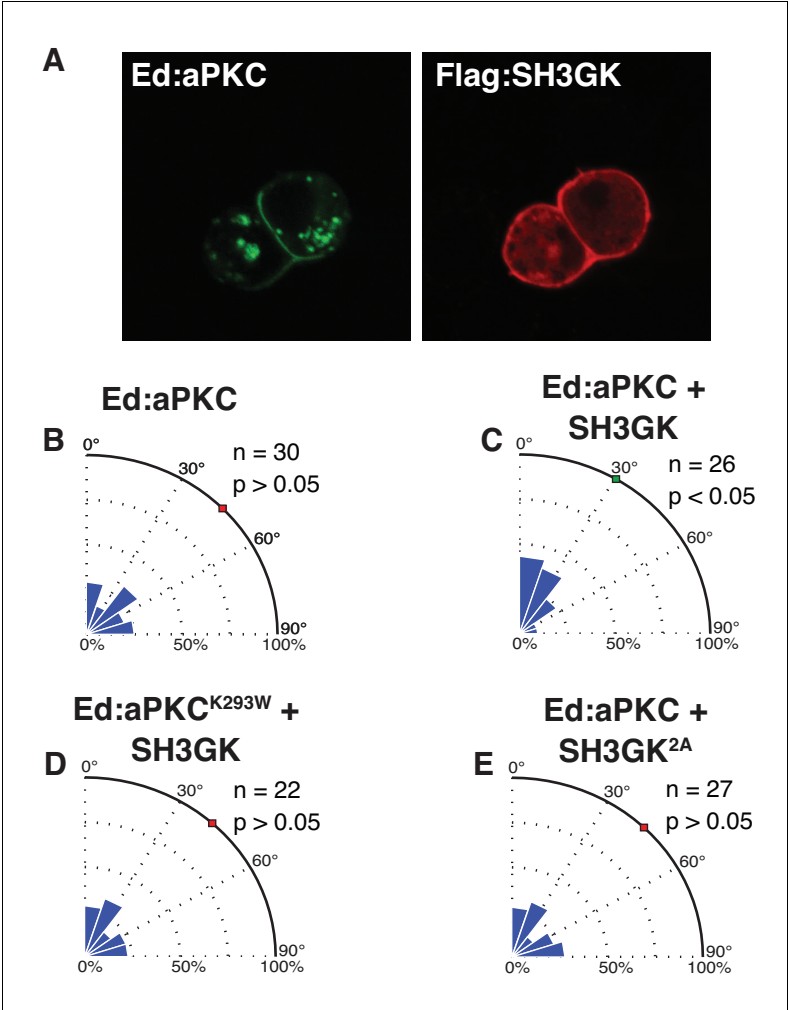

**Figure 4.** Dlg mediates spindle alignment by aPKC. Ed:aPKC was polarized in S2 cells and in some cases Dlg was co-expressed. (**A**) Localization of Ed:aPKC and SH3GK in S2 cells. Spindle measurements were made for the following conditions (**B**) Ed:aPKC (**C**) Ed:aPKC co-expressed with Dlg SH3GK (**D**) Ed:aPKC containing the 'kinase-dead' K293W mutation co-expressed with Dlg SH3GK. (**E**) Ed:aPKC co-expressed with Dlg SH3GK containing alanine substitutions at the two aPKC phosphorylation sites. For each condition, a polar histogram of measured angles is shown along with a p-value calculated by comparison against Ed:GFP (Ed:aPKC) or Ed:aPKC (all others) using the K-S test. Individual measurements are included in comma separated value format in *Figure 4—source data 1*.

DOI: https://doi.org/10.7554/eLife.32137.007

The following source data is available for figure 4:

**Source data 1.** Individual spindle angle measurements in comma separated value format (corresponding figure panel is included in the column header).

DOI: https://doi.org/10.7554/eLife.32137.008

intramolecular interaction prevents GukH recruitment consistent with our observations with purified components (*Figure 5A*). GukH is efficiently recruited; however, to crescents of Ed:SH3GK containing either of the phosphomimetic mutations S662D or S676D, or to wild-type Ed:SH3GK when aPKC is expressed (*Figure 5E–G,J*). Recruitment to Ed:SH3GK by aPKC requires its kinase activity (*Figure 5H,J*) and the presence of phosphorylatable residues at the SH3 sites (*Figure 5I,J*). Taken together with our functional spindle orientation results, these data support a model in which aPKC spatially regulates Dlg spindle orientation activity by allowing it to recruit the GK ligand GukH specifically at the apical cortex where aPKC and Dlg colocalize.

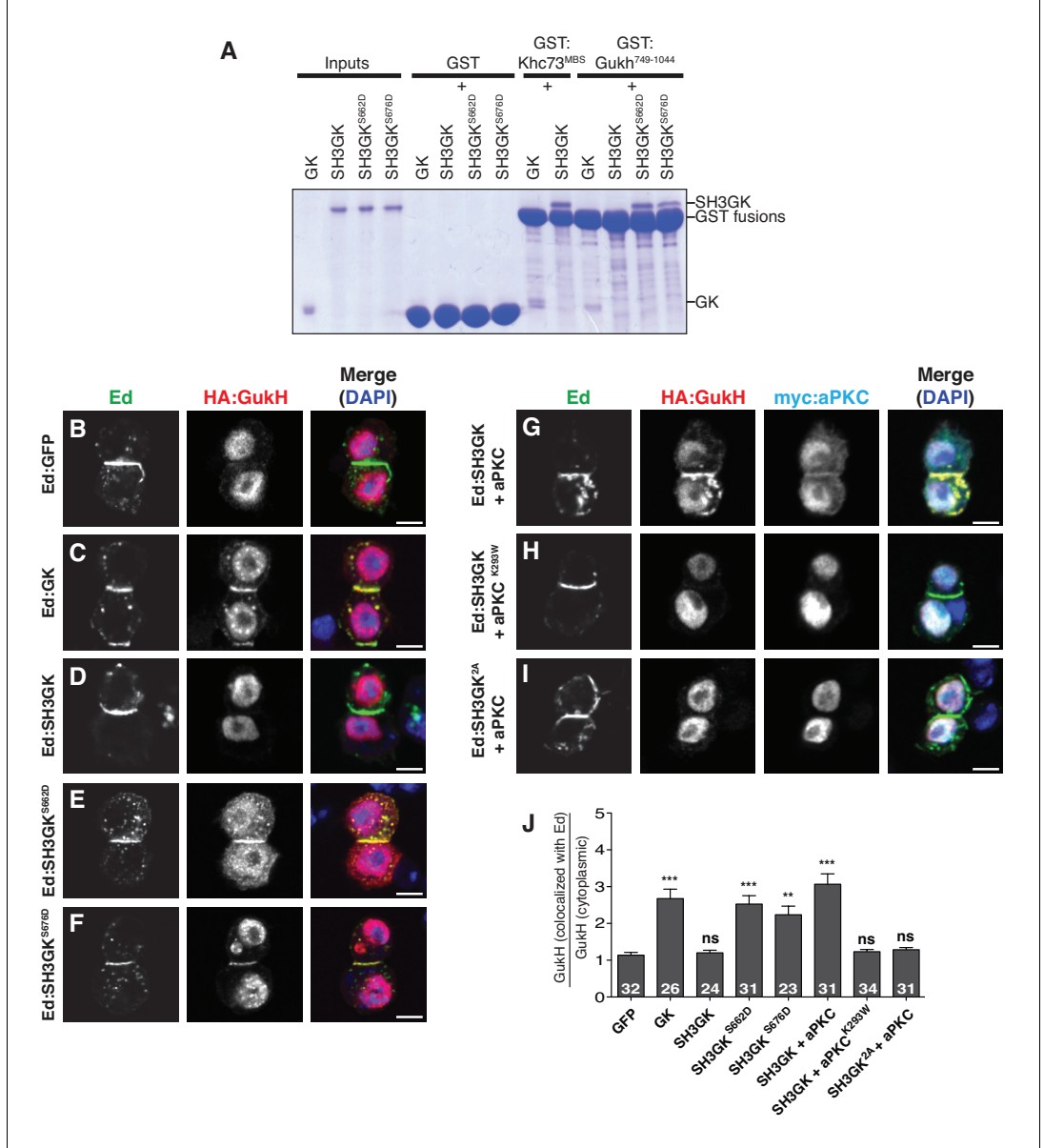

**Figure 5.** aPKC phosphorylation activates GukHolder binding and recruitment by Dlg SH3GK. (**A**) GK versus SH3GK binding of Khc73 and GukHolder. 'Inputs' are approximately 2 μg of purified Dlg GK and SH3GK proteins. 'GST' lanes are the results of affinity chromatography with GST alone to measure the amount of background binding. 'GST-Khc73$^{MBS}$' lanes are the Kinesin Khc73's 'MAGUK Binding Stalk' with the Dlg and GK SH3GK, showing no qualitative effect on binding by the autoinhibitory SH3GK interaction. 'GST-GukH$^{749-1044}$' lanes are the GukH COOH-terminus with Dlg GK and SH3GK. Binding to the GK but not the SH3GK indicates that GukH is inhibited by the SH3GK intramolecular interaction. Binding to SH3GKs containing phosphomimetic mutations suggests that aPKC phosphorylation overcomes autoinhibition. (**B–J**) Cultured S2 cells expressing full-length GukH and the following Ed fusions with Dlg stained for Ed, GukH and DAPI (**B**) Ed fused to GFP (**C**) Ed fused to Dlg GK (**D**) Ed fused to Dlg SH3GK (**E**) Ed:SH3GK with phosphomimetic mutation at S662 (**F**) Ed:SH3GK with phosphomimetic mutation at S676 (**G**) Ed:SH3GK co-expressed with aPKC (**H**) Ed:SH3GK co-expressed with 'kinase-dead' aPKC (**I**) Ed:SH3GK containing alanine mutations at the aPKC phosphorylation sites co-expressed with aPKC (**J**) Quantification of GukH recruitment to Ed-fusion proteins. The number of measurements is shown in each bar and significance (**p<0.01, ***p<0.001) calculated by comparison with the GFP condition calculated using ANOVA with Dunnett's post-test.
DOI: https://doi.org/10.7554/eLife.32137.009

## GukH is a microtubule-binding spindle orientation protein

Although GukH is required for GK-mediated spindle orientation in S2 cells (*Garcia et al., 2014*) and is localized to the apical cortex in metaphase neuroblasts (*Albertson and Doe, 2003*), its

requirement for spindle orientation in vivo had not been determined. We generated a *Drosophila* transgenic line containing a UAS-driven short hairpin RNAi directed against a *gukh* exon found in all predicted splice forms of the gene product. When we expressed the RNAi using the neuroblast-specific *worniu*-Gal4 driver, we observed a spindle orientation defect in larval neuroblasts indistinguishable from that observed in neuroblasts from *dlg* mutants (*Figure 3C* and *6A,B*) (*Albertson and Doe, 2003*; *Siegrist and Doe, 2005*). Thus, GukH function is required for spindle orientation in vivo and the *gukh* phenotype is consistent with its function in the Dlg pathway. Additionally, we find that

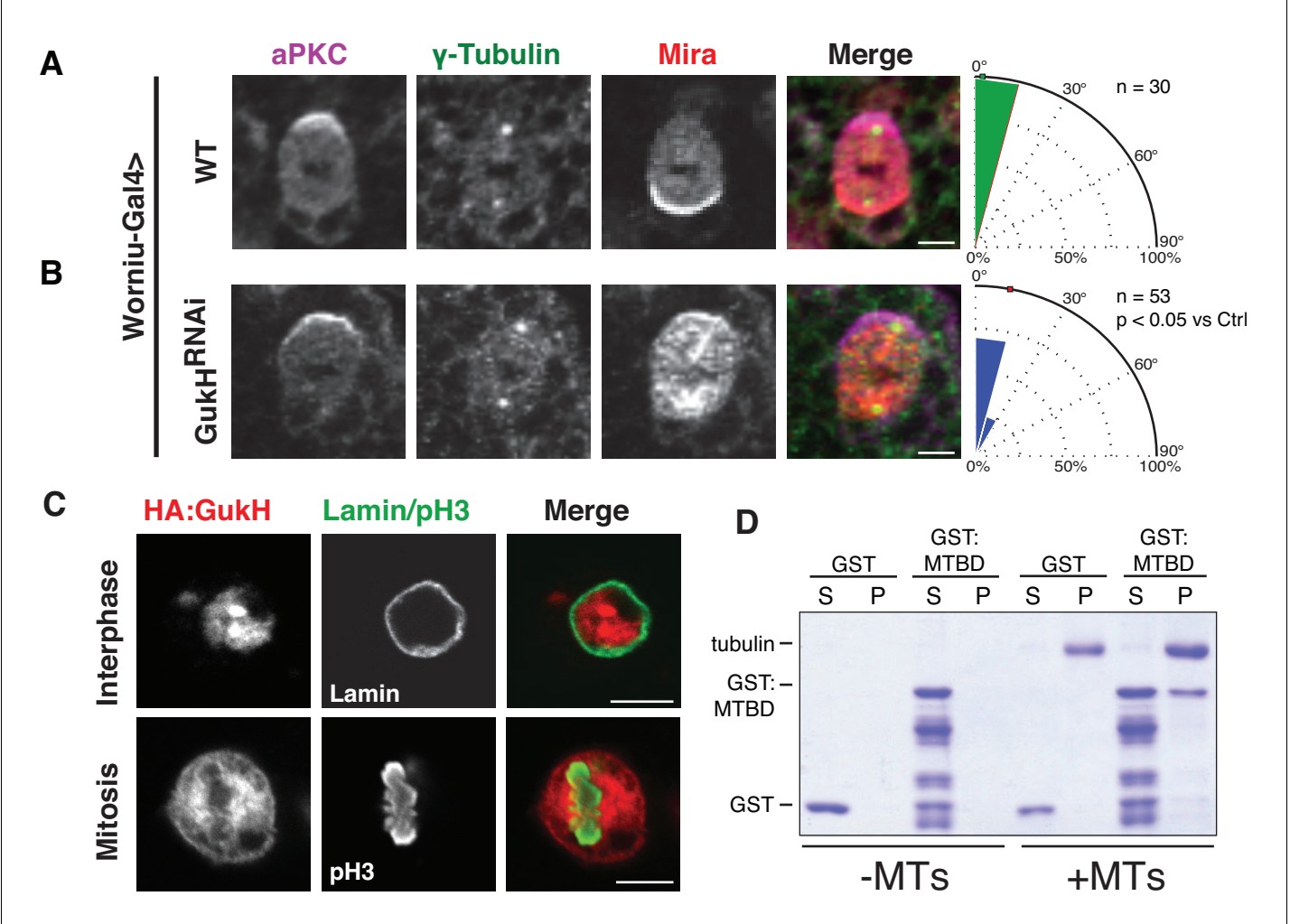

**Figure 6.** Gukholder is a microtubule-binding protein that mediates neuroblast spindle orientation. Spindle angle measurements from larval brain neuroblasts stained for aPKC, γ-tubulin (used to measure spindle angles), and Mira from animals with the following genotypes (A) Worniu-Gal4 >WT (no RNA expressed) (B) Worniu-Gal4 >GukH RNAi. Polar histograms of angle measurements are shown along with the p-value calculated by comparison with the control RNAi condition using the K-S test. (C) Localization of GukH in interphase and mitotic S2 cells. Cells were stained for HA to detect tagged full-length GukH, Lamin for the nuclear membrane, and pH3 for mitotic chromatin. (D) GST and a GST fusion of the putative GukH microtuble-binding domain (MTBD; residues 404-534) were subjected to ultracentrifugation both in the absence (–MTs) and presence (+MTs) of microtubules. The supernatant and pellet from these experiments are shown. Individual measurements are included in comma separated value format in *Figure 6—source data 1*.

DOI: https://doi.org/10.7554/eLife.32137.010

The following source data and figure supplement are available for figure 6:

**Source data 1.** Individual spindle angle measurements in comma separated value format (corresponding figure panel is included in the column header).
DOI: https://doi.org/10.7554/eLife.32137.012
**Figure supplement 1.** GukHolder is not required for localization of its binding partner, the polarity protein and tumor suppressor Scribble.
DOI: https://doi.org/10.7554/eLife.32137.011

GukH plays a role in Dlg's function in cell polarity (*Ohshiro et al., 2000*; *Peng et al., 2000*), as GukH is required for Miranda polarity (*Figure 6A,B*). As GukH is known to bind the tumor suppressor Scribble (Scrib) (*Mathew et al., 2002*), we tested if GukH influences Scrib localization in neuroblasts. We found that Scrib localization is independent of GukH suggesting that either GukH functions downstream of Scrib, or its role in polarity is Scrib-independent (*Figure 6—figure supplement 1*).

Having established that GukH is required for spindle orientation in vivo, we next sought to characterize the mechanism by which it participates in spindle orientation. We noticed that full-length GukH isoform C (GukH$^C$) localizes to the nucleus of interphase S2 cells, but becomes diffusely localized to the spindle and cell cortex during mitosis (*Figure 6C*). GukH's localization on the spindle in this overexpression context led us to hypothesize that it may bind microtubules (MTs). To test this hypothesis, we purified a fragment of GukH$^{404-534}$ that contains a highly positively charged region we predicted to be a putative MT-binding domain (MTBD) and tested its ability to bind microtubules using a MT pelleting assay. Compared to the GST-alone negative control, cosedimentation of GukH$^{MTBD}$ revealed significant pelleting with assembled microtubules (*Figure 6D*). We conclude that GukH is a novel microtubule-binding spindle orientation protein.

## GukH microtubule-binding activity is required for GK-mediated spindle orientation

To determine whether GukH's MT-binding activity directly contributes to Dlg-mediated spindle orientation activity, we tested whether its MTBD is required for spindle orientation in S2 cells. We first verified that intact GukH is required for Ed:GK-mediated spindle orientation in S2 cells (*Figure 7A, B*) by depleting endogenous GukH using RNAi targeted against the *gukh* 5' UTR (*Garcia et al., 2014*). Expression of full-length GukH rescued the loss in activity caused by *gukh*$^{5'UTR}$ RNAi (*Figure 7C*), demonstrating that the RNAi is specific to our gene of interest and that this isoform is sufficient for spindle orientation. Expression of GukH lacking its MTBD did not rescue the *gukh* RNAi spindle orientation phenotype, suggesting that MT-binding activity is essential for GukH to function in the Dlg spindle orientation pathway (*Figure 7D*). We conclude that GukH is a novel MT-binding protein and this activity is required for Dlg-mediated spindle orientation.

## Discussion

The precise alignment of the mitotic spindle to a cortical mark is important for many cellular processes, including oriented divisions and differentiation (*Gönczy, 2008*; *Lu and Johnston, 2013a*; *di Pietro et al., 2016*; *Roubinet and Cabernard, 2014*). The first step in aligning the spindle to the cortex is selecting the cortical site followed by recruitment of protein complexes such as microtubule interacting proteins that will bind to and exert force on astral microtubules. Although the cellular signaling pathways that control cortical site selection ultimately determine where the spindle will be located, we are just beginning to understand the mechanisms that couple signaling systems to spindle orientation complexes. We have identified a mechanism that links the polarity kinase aPKC to an essential spindle orientation factor, GukH via the tumor suppressor Dlg. In previous work (*Newman and Prehoda, 2009*), we found that autoinhibition of Dlg is required for spindle orientation during neuroblast asymmetric cell division, but the precise role of autoinhibition has been unclear. Here, we show that autoinhibition allows Dlg to function as a regulated adaptor that only recruits GukH when it is phosphorylated by aPKC. Regulation takes advantage of an intramolecular interaction between the Dlg SH3 and GK domains, a common feature of MAGUK proteins (*Funke et al., 2005*). This mechanism allows Dlg to be recruited to the cortex uniformly, but to only be activated at cortical sites containing aPKC, which may have implications for both spatial and temporal control of spindle orientation. Altogether, these findings suggest that aPKC promotes spindle alignment through Dlg by: (1) phosphorylating Dlg at the apical cell cortex, (2) which prevents SH3-mediated inhibition of GK ligand binding and (3) recruits the downstream spindle orientation protein GukH which (4) establishes a connection to the mitotic spindle by directly binding to microtubules exclusively at the apical cell cortex (*Figure 7E*). It is important to note that while our work highlights Dlg's role in connecting aPKC and GukH, it does not exclude a role for other Dlg effectors such as Khc73 (*Lu and Prehoda, 2013b*) or Banderuola (*Mauri et al., 2014*) in spindle orientation.

An important implication of our findings is the identification of a new molecular connection between the cortically polarized Par complex and the mitotic spindle. Currently, the best-

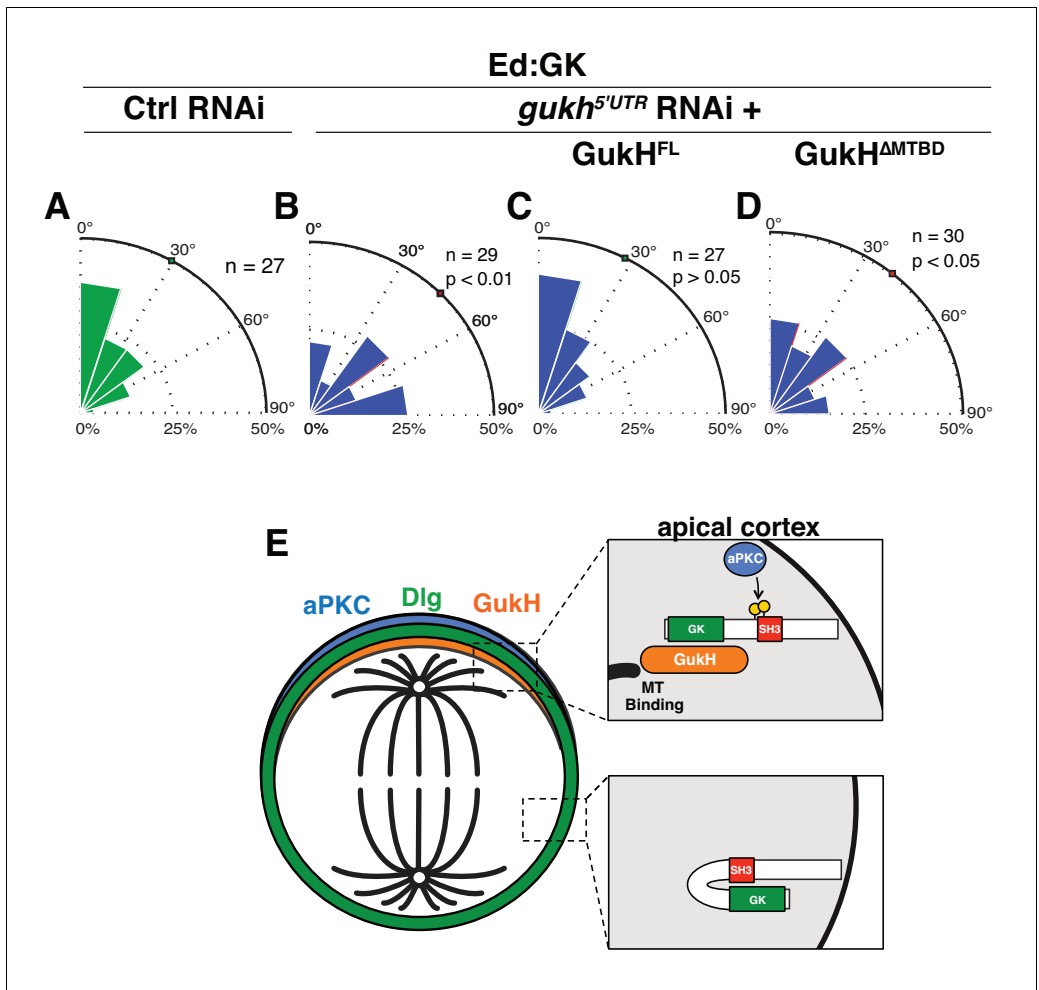

**Figure 7.** The GukHolder microtubule binding domain mediates spindle orientation. Spindle angles measured in S2 cells expressing Ed fused to Dlg GK exposed to the following RNAi conditions (**A**) Control RNAi (**B**) RNAi directed against the *gukh* 5' UTR (**C**) *gukh*$^{5'UTR}$ RNAi with expression of full-length GukH (**D**) *gukh*$^{5'UTR}$ RNAi with expression of full-length GukH lacking its microtuble-binding domain (MTBD). For each condition, a polar histogram of the measured angles is shown and a p-value calculated by comparison to the control RNAi condition using the K-S test. (**E**) Model for coupling of polarity and spindle orientation during asymmetric cell division. When Dlg is not co-localized with aPKC, it is autoinhibited by the SH3GK intramolecular interaction and cannot recruit GukH. Phosphorylation of Dlg by aPKC relieves autoinhibition allowing it to interact with GukH and recruit it to sites containing aPKC. The GukH interaction with microtubules mediates spindle orientation. Individual measurements are included in comma separated value format in *Figure 7—source data 1*.

DOI: https://doi.org/10.7554/eLife.32137.013

The following source data is available for figure 7:

**Source data 1.** Individual spindle angle measurements in comma separated value format (corresponding figure panel is included in the column header).

DOI: https://doi.org/10.7554/eLife.32137.014

characterized link between the Par complex and spindle orientation is the scaffolding protein Inscuteable (*Knoblich, 2010*; *Schober et al., 1999*; *Wodarz et al., 1999*). As Inscuteable binds directly to the Par complex member Bazooka (Baz; aka Par-3) and Partner of Inscuteable (Pins; aka LGN), expression of Inscuteable in epithelia and neural stem cells redirects spindle orientation toward the Par complex by directly recruiting Pins and associated spindle orientation proteins such as Mushroom Body Defect (Mud; aka NuMA). Our data demonstrate that aPKC provides an additional link between the Par complex and spindle orientation complexes through phosphorylation and activation of Dlg. In light of biochemical and structural data that Inscuteable and Mud form

mutually exclusive complexes with Pins (*Mauser and Prehoda, 2012*; *Zhu et al., 2011*), we propose that aPKC-Dlg signaling provides an essential cue from the apical cortex to promote alignment of the mitotic spindle along the apical-basal axis.

An important prediction of this model is the requirement of aPKC for spindle orientation. In epithelial tissues, such as MDCK epithelial cells, the *Drosophila* imaginal wing disc epithelium, and Caco-2 epithelial 3D cysts (*Durgan et al., 2011*; *Guilgur et al., 2012*; *Hao et al., 2010*), aPKC is required for spindle alignment within the plane of the epithelium, although it does not appear to be essential for spindle orientation in chick neuroepithelial cells or the *Drosophila* follicle cell epithelium (*Bergstralh et al., 2013*; *Peyre et al., 2011*) but is in neuroblasts and the embryonic epithelium (*Guilgur et al., 2012*; *Kim et al., 2009*). These results suggest that the aPKC function in spindle orientation may be cell specific. Dlg is required for planar spindle alignment in the *Drosophila* follicle cell epithelium (*Bergstralh et al., 2013*), but the requirement of Dlg in other tissues has not been investigated.

In addition to unveiling a new link between the Par complex and mitotic spindle orientation proteins, we also identified a regulatory function for Dlg interdomain interactions in mitotic spindle orientation. The SH3GK interaction allosterically regulates GK ligand binding of Dlg and other members of the MAGUK family, although the precise function of this regulatory interaction has not been known. Our work highlights the importance of modulating Dlg interdomain interactions through phosphoregulation and is consistent with previous work showing a cellular role for aPKC-Dlg signaling. A mammalian PKC isoform, PKCα, phosphorylates human Dlg1 (hDlg1) at a threonine residue within the Hook domain (T656) and regulates primary astrocyte migration (*O'Neill et al., 2011*). The homologous residue to human Dlg T656 in *Drosophila* Dlg is S676 that we show is phosphorylated by aPKC, suggesting that phosphorylation of Dlg within the Hook domain by aPKC is an evolutionarily conserved event. Furthermore, the mammalian aPKC homolog, PKCζ, is required for Dlg1 localization to the primary astrocyte leading edge where it binds to the GK-ligand GKAP, which recruits dynein and regulates centrosome positioning during migration (*Etienne-Manneville et al., 2005*; *Manneville et al., 2010*). Similar to how aPKC-Dlg signaling restricts GK-binding capacity to the apical cortex in mitotic neuroblasts, PKC phosphorylation of hDlg1 could restrict GK-binding capacity to the leading edge of primary astrocytes, which suggests that Dlg domain allostery can be spatially controlled by phosphoregulation to produce a specific cellular response in diverse cell types.

## Materials and methods

### Cloning, protein expression and purification

Dlg PDZSH3GK (residues 474–975) and SH3GK (residues 598–975) fragments were cloned into the pGEX vector for N-terminal GST fusions and the pBH vector for an N-terminal His$_6$ tag. pGEX and pBH vectors were transformed into BL21 *Escherichia coli* cells and protein expression was induced by addition of 0.5 mM IPTG at 16°C overnight. Bacterial lysates containing GST fusions were used in qualitative pulldown assays. For His-tag purification, bacterial lysates were incubated with Ni$^{2+}$-nitrilotriacetic acid resin at 4°C for 2 hr. The resin was washed with lysis buffer, eluted with Ni$^{2+}$ elution buffer (50 mM NaH$_3$PO$_4$, 300 mM NaCl, 250 mM imidazole) and dialyzed overnight at 4°C.

Purification of His-tagged Gukholder (residues 749–1044) was performed as previously described for aPKC (*Graybill et al., 2012*). Briefly, HEK293T cells were transfected with His$_6$-tagged Gukholder cloned into the pCMV vector. The cells were incubated for 72 hr at 37°C. Lysates were obtained by sonication of cell cultures followed by centrifugation at 15,000 rpm for 30 min at 4°C. 45% (w/v) ammonium sulfate was added to lysates and the resulting precipitant was centrifuged at 15,000 rpm for 30 min at 4°C and the pellet was resuspended in Ni$^{2+}$ lysis buffer (50 mM NaH$_3$PO$_4$, 300 mM NaCl, 10 mM imidazole). Resuspended pellets were incubated with Ni$^{2+}$-nitrilotriacetic acid resin at 4°C for 45 min. The resin was washed, eluted, and dialyzed as above.

Full length Gukh$^C$ cDNA was cloned from an embryonic/larval cDNA library prepared from Oregon-R flies using TRIzol reagent (Thermo Fisher Scientific). Primers used were F – 5′ cacacctaggA TGCCCTTCGTCCAGCGCGTGGTGCAG 3′ and R – 5′ atagtttagcggccgcCTACAACGCCG TTTCAAACGAGGGCAG 3′ containing 5′ AvrII and 3′ NotI overhangs for restriction cloning. The PCR product was gel purified and ligated into a modified pMT-HA backbone containing a custom

MCS for use with AvrII/NotI. Attempts to amplify isoforms A and E were unsuccessful, suggesting those isoforms may be expressed during later stages of development.

## In vitro kinase assay

aPKC phosphorylation of Dlg was performed as previously described (*Graybill et al., 2012*). Briefly, 20 µM substrate was pre-incubated with purified aPKC kinase domain in reaction buffer (20 mM HEPES, pH 7.5, 10 mM MgCl$_2$, 1 mM DTT) at 30°C for 5 min. Reactions were initiated by addition of 1 mM ATP and incubated at 30°C for 25 min. Reactions were quenched by boiling and analyzed by mass spectrometry (Fred Hutchison Cancer Center Proteomics Core).

## In vitro binding assay

GST fusions of Dlg fragments and mutants were incubated with glutathione agarose beads at 4°C for 30 min and subsequently washed three times with binding buffer (10 mM HEPES, pH 7.5, 100 mM NaCl, 1 mM dithiothreitol, 0.002% Triton X-100). His-tagged Gukholder (amino acids 749–1044) was added to washed beads and incubated at 4°C for 1 hr. The reactions were then washed three times with binding buffer, eluted by addition of SDS loading buffer, and ran on a 12.5% SDS poly-acrylamide gel. SDS-PAGE gels were transferred onto nitrocellulose and blocked in 5% skim milk in TBS-T. Blots were subjected to Western analysis by probing with a primary mouse anti-His antibody (1:1000; Santa Cruz Biotechnology, Santa Cruz CA) and IRDye 800CW goat anti-mouse secondary antibodies and visualized using LiCOR Odyssey.

## MT pelleting assay

Recombinant GST:Gukh fusion proteins were expressed in *E. coli* BL21-DE(3) cells and purified by binding to glutathione agarose beads, rinsed three times with wash buffer pH 7.4 1 mM, DTT in PBS and eluted with GST elution buffer 10 mM HEPES pH 7.5, 100 mM NaCl, 1 mM DTT. The Gukh microtubule binding domain, Gukh$^{MTBD}$, corresponds to amino acids 404–534 in the Gukholder iso-form C sequence, and Gukh$^{MTBD-\Delta PBM}$ corresponds to deletion of amino acids 500–512 from the MTBD. Purified proteins were dialyzed in Salt-free Dialysis Buffer, pH 7.4 25 mM Hepes 110 mM KOAc, 20 mM imidazole, 1 mM MgCl$_2$,1 mM EGTA, 0.02% Tween, 5% sucrose, 1 mM DTT and con-centrated using a Sartorius VivaSpin 3,000 MW cut-off spin column, and verified for purity and con-centration on SDS-PAGE gel. Microtubule binding and pelleting assay was performed as directed in Cytoskeleton Inc. Microtubule Binding Protein Spin Down Assay, Catalog Number: BK029, Lot num-ber 121. Microtubule binding was assessed by diagnostic SDS-PAGE gel.

## S2 cell culture, transfection, and staining

S2 cells were cultured, transfected, stained and visualized as described previously (*Wee et al., 2011*). Briefly, Dlg fragments and mutants or aPKC were cloned downstream of Ed:GFP or Ed:tdTom in the pMT vector. Flag-tagged Dlg fragments and aPKC were cloned with an N-terminal Flag tag into the pMT vector. *Drosophila* Schneider's (S2) cells were maintained in Schneider's medium (Invitrogen, Carlsbad CA) with 10% heat-inactivated fetal bovine serum (Cellgro; Mediatech, Manassas VA) at room temperature. For transfection of S2 cells, approximately $3 \times 10^6$ cells were plated per well and transfected with 1 µg total DNA according to the Effectene manufacturer's protocol (Qiagen, Venlo, Netherlands). Transfection complexes were incubated over-night and protein expression was induced by addition of 0.5 mM CuSO$_4$ for 24 hr. For RNAi experi-ments, $1 \times 10^6$ cells were transferred to serum-free media before addition of dsRNA, allowed to incubate for 1 hr, and serum-containing media was added to the cells to induce growth. RNAi was performed for 72 hr. Cells were then collected, resuspended in fresh growth medium, and shaken at 175 rpm for 2–3 hr to induce localized clustering of Ed transfected cells. 200 µL of clustered cells were plated on 12 mm diameter coverslips in 24-well plates, allowed to adhere for 1 hr, followed by addition of 300 µL of fresh growth medium for 3–4 hr. Coverslips were fixed in 4% paraformalde-hyde (PFA) in PBS, washed three times in washing buffer (0.1% saponin in PBS), and blocked in blocking buffer (0.1% saponin, 1% bovine serum albumin (BSA) in PBS). Primary antibodies used were mouse anti-αtubulin (1:1000; Santa Cruz Biotechnology), mouse anti-HA (1:1000; Covance, Denver PA), rat anti-tubulin (1:1000; Pierce Biotechnology, Rockford IL), mouse anti-Lamin (1:1000; DSHB), rabbit anti-pH3 (1:7500; Millipore, Billerica MA), mouse anti-myc (1:1000;

Sigma Aldrich), and mouse anti-Flag (1:1000; Sigma Aldrich). Coverslips were incubated with primary antibodies overnight at 4°C, washed in washing buffer, incubated with fluorophore-conjugated secondary antibodies (1:500; Jackson Immunoresearch, West Grove PA) in blocking buffer for 2 hr at room temperature, rinsed three times with washing buffer, and mounted on microscope slides using Vectashield Hardset. S2 cell images were acquired using either a Bio-Rad Radiance 2100 confocal microscope using a 1.4NA 60x oil objective, or an Olympus FV1000 confocal microscope using a 1.42NA 60x oil immersion objective. All images were quantified and processed using FIJI (ImageJ), and figures were assembled in GraphPad PRISM 5.0 and Adobe Illustrator CS6.

### *Drosophila* genetics and brain dissections

Stocks used were as follows: w1118 (Bloomington Stock Center) was used as WT; y,v;;UAS-Gukh$^{RNAi}$, homozygous short hairpin insertion on chromosome III was generated by ligating a F/R primer cassette with the sequence: 5'ctagcagtTCCAGCGATCACGAAGTTCTAtagttatattcaagcataTAGAACTTCGTGATCGCTGGAgcg 3' into pVALIUM20 (TRiP protocol) and injecting plasmid into y,v;;attP2 flies (Genetic Services); y,v;;attP2 was used as the control RNAi; dlg$^{m52}$/FM7i,B,Act:GFP;Wor-Gal4/Cyo,Act:GFP (gift from Chris Doe lab); FM7i,B,Act:GFP/Y, hemizygous males containing Act:GFP labeled X chromosome for Ctrl rescue selection; FM7i,B,Act:GFP/Y;;UAS-emGFP:Dlg$^{WT}$, homozygous transgene insertion on chromosome III; FM7i,B,Act:GFP;;UAS-emGFP:Dlg$^{S662D}$, homozygous transgene insertion on chromosome III. Gukh RNAi was performed by driving UAS-shRNA expression with the Wor-Gal4 driver at 29°C. Brains were harvested in third instar larval stages. Dlg deficient third instar larvae were obtained by crossing dlg$^{m52}$/FM7i,B,Act:GFP; Wor-Gal4/Cyo,Act:GFP females with FM7i,B,Act:GFP males and screening against Act:GFP. The resulting larvae were all GFP$^-$ males of genotype dlg$^{m52}$/Y;Wor-Gal4/+;+/+. Note: dlg$^{m52}$ hemizygous male larvae die at the larval-pupal transition and exhibit small testes that render them indistinguishable from females. Rescue crosses were performed by crossing dlg$^{m52}$/FM7i,B,Act:GFP; Wor-Gal4/Cyo,Act:GFP females with transgenic FM7i,B,Act:GFP/Y;;UAS-emGFP:Dlg homozygous males and screening against Act:GFP. The resulting larvae were all GFP$^-$ males of genotype dlg$^{m52}$/Y;Wor-Gal4/+;UAS/+.

All neuroblasts shown are from 3$^{rd}$ instar larval stage when maternal Dlg load is diminished in neuroblasts. Larval brains were dissected in Schneider's Media without serum, followed by a 20-min fixation in PBS with 4% paraformaldehyde. Brains were permeabilized with 0.3% Triton-X100 in PBS for 30 min, then blocked in 0.3% Triton-X100% and 1% BSA for 1 hr at room temperature. Primary antibodies were diluted in blocking buffer and incubated overnight at 4°C.

Primary antibodies used were mouse anti-Dlg (1:1000; DSHB), rabbit anti-PKCζ (1:1000; Santa Cruz Biotechnology); rabbit anti-Cnn (1:500; gift from Kaufman Lab), rat anti-Mira (1:1000; Abcam, Cambridge MA), mouse anti-γtubulin (1:500; Sigma Aldrich). Brains were washed 3x with block solution. Secondary antibodies were diluted in blocking buffer but were incubated for 1.5 hr at room temperature.

Secondary antibodies used were donkey anti-mouse Alexa-488, donkey anti-rabbit DyLight-649, donkey anti-rabbit Alexa-647, donkey anti-rat Cy3, all obtained form Jackson Immunoresearch. All secondary antibodies were highly cross-adsorbed against other species in order to minimize cross-reactivity and false-positive signal. Samples were thoroughly washed in PBS then stored and mounted in SlowFade Diamond with DAPI solution (Thermo Fisher Scientific).

Brain scans were performed using either an Olympus FV1000 laser scanning confocal using a 1.3NA 40x oil objective, or a Zeiss LSM880 laser scanning confocal microscope equipped with a S.1 super resolution structured illumination module using a 1.4NA 40x oil objective. Images were quantified and processed using FIJI (ImageJ) and figures assembled in GraphPad PRISM 5.0 and Adobe Illustrator CS6.

## Additional information

### Funding

| Funder | Grant reference number | Author |
| --- | --- | --- |
| National Institutes of Health | GM087457 | Kenneth E Prehoda |

The funders had no role in study design, data collection and interpretation, or the decision to submit the work for publication.

## Author contributions
Ognjen Golub, Conceptualization, Formal analysis, Investigation, Methodology, Writing—review and editing; Brett Wee, Conceptualization, Investigation, Methodology, Writing—original draft, Writing—review and editing; Rhonda A Newman, Conceptualization, Investigation, Writing—review and editing; Nicole M Paterson, Investigation, Writing—review and editing; Kenneth E Prehoda, Conceptualization, Formal analysis, Supervision, Funding acquisition, Writing—original draft, Project administration, Writing—review and editing

## Author ORCIDs
Kenneth E Prehoda (iD) http://orcid.org/0000-0003-4214-6158

## Decision letter and Author response
Decision letter https://doi.org/10.7554/eLife.32137.017
Author response https://doi.org/10.7554/eLife.32137.018

# Additional files
## Supplementary files
• Transparent reporting form
DOI: https://doi.org/10.7554/eLife.32137.015

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
