## [Decision Letter]

Thank you for submitting your article "Activation of Discs large by aPKC couples cell polarity and the mitotic spindle during asymmetric cell division" for consideration by *eLife*. Your article has been favorably evaluated by K VijayRaghavan (Senior Editor) and two reviewers, one of whom, Yukiko M Yamashita (Reviewer #1), is a member of our Board of Reviewing Editors.

The reviewers have discussed the reviews with one another and the Reviewing Editor has drafted this decision to help you prepare a revised submission.

Summary:

This manuscript describes a mechanism by which spindle orientation is connected to cortical polarity. Specifically, the authors show that Dlg (cortical polarity) is connected to microtubule via GukH, and their interaction is regulated by aPKC-mediated phosphorylation. The study is conducted to a high standard, and the manuscript was reviewed favorably.

Essential revisions:

Whereas the study is conducted to a highest standard, one of the reviewers raised a question as to whether the study provides sufficient progress compared to the existing knowledge in the field, in particular in the light of the authors' own study concerning Dlg and GukH (Newman and Prehoda 2009). (We understand that the previous study did not show GukH's direct interaction with microtubule and thus the present study represents a good advancement in showing directness of the interaction.) With that said, it is important that the manuscript precisely conveys what is the major advancement of the present study, and we ask the authors to revise the manuscript accordingly.

In addition, the reviewers felt that the manuscript can benefit by addressing some outstanding questions. For example:

1) Can the authors delete the Guk domain and see if it recapitulates the dlg[M52] mutant phenotype? If Guk domain is essential for Dlg function, such experiments will reveal the key mechanism by which Dlg regulates asymmetric neuroblast division.

2) Why over-expressing Dlg[S662D] only rescued mis-alignment of the spindle in ~60% of neuroblasts whereas the full-length completely rescued this phenotype?

3) Mauri et al., proposed Bnd functions in parallel to Dlg. 'They probably should take this paper into consideration, and somehow integrate it into their work to broaden the scope of this study.

Reviewer #1:

In this manuscript, Prehoda and colleagues study how spindle orientation is linked to cell polarity, a critically important question in understanding the mechanism of asymmetric cell divisions. They provide an elegant mechanism by which Dlg is regulated by the activity of aPKC to anchor (or not) GukH, which in turn orients spindle.

This is a very careful and thorough study that provides a critical new link between cell polarity and spindle orientation. This is an important study that adds to our knowledge in spindle orientation, and I am in favor of publishing this manuscript in *eLife*.

1) GukH localization in neuroblast? Especially because its localization in S2 cells is not so convincing, I'd love to see its localization in neuroblasts.

2) Figure 6: why does Miranda mislocalize upon GukH RNAi? I might be missing some well-established relationship between apical and basal crescent in neuroblasts, but I thought that disruption of spindle orientation pathway (instead of 'apical polarity pathway') wouldn't cause basal crescent disruption. Even if it's a well-known phenomenon, elaborating it would help accessibility of general readers. Also, in this figure, aPKC is the only apical protein that was examined. It may help understanding of GukH-mediated spindle orientation mechanism if they check other apical proteins.

3) Spindle orientation phenotype of dlg and GukH mutants/RNAi seems to be somewhat weak. Certainly, it can be redundancy with other pathways, such as Insc – it might be helpful at least to discuss about this point.

Reviewer #2:

The manuscript by Golub et al. explores how spindle orientation is coupled with cortical cell polarity, and insights into how these two processes are precisely coordinated are directly relevant to virtually all biological contexts involving asymmetric cell divisions. Disc large (Dlg) is a classical neoplastic tumor suppressor gene that was initially identified in flies and was subsequently characterized in other vertebrate contexts. A role for Dlg in regulating spindle orientation is well established, but the mechanisms are not understood. Furthermore, how Dlg functions coordinately with cortical polarity cues mainly the Par complex to regulate asymmetric cell division was also poorly understood. This study offers new insights indicating that phosphorylation by aPKC contributes to the activation of Dlg functions in orienting the mitotic spindle. The reviewer appreciates the multidisciplinary approaches used in this study to unravel the mechanism.

Although the data are solid and the experiments are well executed, the findings of this study appear extremely narrow, and do not significantly improve our understanding of how cortical cell polarity and spindle orientation are precisely coordinated. This is in part due to an earlier work from the same lab (Newman and Prehoda, 2009) provided some evidence suggesting that the intramolecular changes likely play an importantly role in Dlg functions. A more profound concern in my mind is the significance of this intramolecular interaction in regulating asymmetric cell division in light of another paper by Mauri et al., 2014 reporting that the dlg[sw] mutant brains do not display ectopic neuroblast formation. This study raised the question of how functional significance of the Guk domain to Dlg function. There are clearly multiple parallel mechanisms that are in place to ensure precise coordination of spindle orientation and asymmetric segregation of various membrane-localized proteins. I am having a really hard time appreciating how these new data improve the overall understanding of the regulation of spindle orientation and asymmetric protein segregation.

1) I would like to suggest that these investigators broaden the breadth of their study by connecting their findings with other parallel mechanisms that coordinate spindle orientation and asymmetric protein segregation. For example, how does the protein Bnd described in Mauri et al., fit into the mechanisms that these investigators describe in this manuscript? How do Khc73 and the dynein-dynactin pathway fit into their model?

2) The authors concluded that Dlg regulates spindle orientation, but the images that they showed in most of their figures lacked some sort of Tubulin staining to show spindle orientation.

3) Why didn't the over-expression of Dlg[S662D] rescue the spindle orientation defects as well as polarization of Miranda to the same extend as Dlg[wt] in Figure 3?

---

## [Author Response]

Essential revisions:Whereas the study is conducted to a highest standard, one of the reviewers raised a question as to whether the study provides sufficient progress compared to the existing knowledge in the field, in particular in the light of the authors' own study concerning Dlg and GukH (Newman and Prehoda 2009). (We understand that the previous study did not show GukH's direct interaction with microtubule and thus the present study represents a good advancement in showing directness of the interaction.) With that said, it is important that the manuscript precisely conveys what is the major advancement of the present study, and we ask the authors to revise the manuscript accordingly.

We appreciate the reviewers’ suggestion to more clearly convey the advance our work represents, and we have modified the text accordingly. In addition to those revisions, we provide a more specific explanation of how the current manuscript advances our understanding over our previous publication (Newman and Prehoda 2009) below. To summarize:

The previous work did not provide a connection between polarity and Dlg-mediated spindle orientation. Our current work demonstrates that the Par polarity kinase aPKC directly modifies Dlg and activates its spindle orientation activity;

The previous work did not provide a connection to a downstream effector that was known to mediate spindle orientation. There are two key points here: 1) other Dlg effectors (such as the kinesin Khc73) were thought to be the key mediators of Dlg spindle orientation and 2) GukHolder was not known to mediate spindle orientation during asymmetric cell division;

The previous work did not provide a function for GukHolder (it wasn’t even known to be a spindle orientation protein and wasn’t known to bind microtubules).

Our current work addresses the fundamental question of how polarity and Dlg-mediated spindle orientation are linked. Newman and Prehoda 2009 did not answer this question. It only showed that Dlg autoinhibition is required for spindle orientation and to repress GukHolder binding, but did not show why autoinhibition is required. It did not provide any mechanism for activating Dlg (which is important because that’s the physical link between polarity and Dlg-mediated spindle orientation), nor did it show that GukHolder was itself involved in spindle orientation. By raising the question as to why Dlg autoinhibition is required for spindle orientation, this previous work did lead us to our current study that provides a complete connection between polarity (Par complex kinase aPKC) and microtubule interactor (discovered in our current manuscript). Thus, our previous work did not answer how polarity and Dlg-mediated spindle orientation are connected – an essential part of asymmetric cell division and the key advance of our current manuscript.

In addition, the reviewers felt that the manuscript can benefit by addressing some outstanding questions. For example:1) Can the authors delete the Guk domain and see if it recapitulates the dlg[M52] mutant phenotype?

Siegriest and Doe 2005 found that the Guk domain is require for neuroblast spindle orientation and we have revised the text to highlight this fact.

If Guk domain is essential for Dlg function, such experiments will reveal the key mechanism by which Dlg regulates asymmetric neuroblast division.2) Why over-expressing Dlg[S662D] only rescued mis-alignment of the spindle in ~60% of neuroblasts whereas the full-length completely rescued this phenotype?

Since Dlg is localized uniformly cortically, the S662D mutation leads to ectopic Dlg activity. In other words, wild type neuroblasts only have active Dlg at the apical cortex because that is where it is phosphorylated by aPKC. Neuroblasts expressing Dlg[S662D], however, have Dlg activity all over the cortex, even at sites lacking aPKC, leading to the spindle misalignment phenotype we observed. We have revised the text to make this phenotype and interpretation more clear.

3) Mauri et al., proposed Bnd functions in parallel to Dlg. 'They probably should take this paper into consideration, and somehow integrate it into their work to broaden the scope of this study.

Our work demonstrates that aPKC phosphorylation of Dlg relieves Dlg autoinhibition, allowing it to bind GukHolder, a spindle orientation protein. Several other Dlg-binding proteins are known to be involved in spindle orientation, including Khc73 and Bnd, and we did not mean to imply that the pathway we have discovered operates at the expense of these other proteins. We have added text to the discussion to clarify this point and thank the reviewer for raising it.

Reviewer #1:In this manuscript, Prehoda and colleagues study how spindle orientation is linked to cell polarity, a critically important question in understanding the mechanism of asymmetric cell divisions. They provide an elegant mechanism by which Dlg is regulated by the activity of aPKC to anchor (or not) GukH, which in turn orients spindle.This is a very careful and thorough study that provides a critical new link between cell polarity and spindle orientation. This is an important study that adds to our knowledge in spindle orientation, and I am in favor of publishing this manuscript in eLife.1) GukH localization in neuroblast? Especially because its localization in S2 cells is not so convincing, I'd love to see its localization in neuroblasts.

Figure 1 in Albertson and Doe (2003) shows clear apical localization of Gukholder in neuroblasts, consistent with our model, but we agree with the reviewer that our own staining of GukHolder would be useful. Unfortunately our attempts to do so have not been successful – we have not been able to obtain any of the antibody used in that original study (reported in Mathew et al. 2002) and our efforts to generate an anti-GukH antibody have not yielded one that works for immunohistochemistry.

2) Figure 6: why does Miranda mislocalize upon GukH RNAi? I might be missing some well-established relationship between apical and basal crescent in neuroblasts, but I thought that disruption of spindle orientation pathway (instead of 'apical polarity pathway') wouldn't cause basal crescent disruption. Even if it's a well-known phenomenon, elaborating it would help accessibility of general readers. Also, in this figure, aPKC is the only apical protein that was examined. It may help understanding of GukH-mediated spindle orientation mechanism if they check other apical proteins.

Besides its role in spindle orientation, Dlg is also known to regulate neuroblast polarity (Ohshiro et al. 2000; Peng et al. 2000; Newman and Prehoda 2009) and our results indicate that GukH is likely to be involved in both functions. To gain additional insight into GukH function we stained GukH RNAi neuroblasts for the tumor suppressor Scribble, a known polarity regulator and

GukH interactor. We found that Scrib localization is unaffected suggesting that it is upstream of GukH’s function. In the revised text we more thoroughly explain the polarity defect and include the new data in Figure 6—figure supplement 1.

3) Spindle orientation phenotype of dlg and GukH mutants/RNAi seems to be somewhat weak. Certainly, it can be redundancy with other pathways, such as Insc – it might be helpful at least to discuss about this point.

As the reviewer notes, Dlg is known to have a mild spindle orientation defect in neuroblasts because of redundancy with the Inscuteable pathway. We emphasize this point in the revised manuscript.

Reviewer #2:The manuscript by Golub et al. explores how spindle orientation is coupled with cortical cell polarity, and insights into how these two processes are precisely coordinated are directly relevant to virtually all biological contexts involving asymmetric cell divisions. Disc large (Dlg) is a classical neoplastic tumor suppressor gene that was initially identified in flies and was subsequently characterized in other vertebrate contexts. A role for Dlg in regulating spindle orientation is well established, but the mechanisms are not understood. Furthermore, how Dlg functions coordinately with cortical polarity cues mainly the Par complex to regulate asymmetric cell division was also poorly understood. This study offers new insights indicating that phosphorylation by aPKC contributes to the activation of Dlg functions in orienting the mitotic spindle. The reviewer appreciates the multidisciplinary approaches used in this study to unravel the mechanism.Although the data are solid and the experiments are well executed, the findings of this study appear extremely narrow, and do not significantly improve our understanding of how cortical cell polarity and spindle orientation are precisely coordinated. This is in part due to an earlier work from the same lab (Newman and Prehoda, 2009) provided some evidence suggesting that the intramolecular changes likely play an importantly role in Dlg functions. A more profound concern in my mind is the significance of this intramolecular interaction in regulating asymmetric cell division in light of another paper by Mauri et al., 2014 reporting that the dlg[sw] mutant brains do not display ectopic neuroblast formation. This study raised the question of how functional significance of the Guk domain to Dlg function. There are clearly multiple parallel mechanisms that are in place to ensure precise coordination of spindle orientation and asymmetric segregation of various membrane-localized proteins. I am having a really hard time appreciating how these new data improve the overall understanding of the regulation of spindle orientation and asymmetric protein segregation.

We appreciate the reviewer’s concerns and have modified the text to more clearly state the advance of our current work (also elaborated on in the text above) and to include the work of Mauri et al.

1) I would like to suggest that these investigators broaden the breadth of their study by connecting their findings with other parallel mechanisms that coordinate spindle orientation and asymmetric protein segregation. For example, how does the protein Bnd described in Mauri et al., fit into the mechanisms that these investigators describe in this manuscript? How do Khc73 and the dynein-dynactin pathway fit into their model?

The advance of our work is to identify a pathway that connects a polarity protein (aPKC) to a spindle orientation protein (GukH) via the tumor suppressor Dlg. We did not mean to imply that other Dlg interacting proteins, including ones that we have worked with (e.g. Khc73) are not essential to this process (although we do show that Khc73 binding to Dlg is not regulated by aPKC). We have revised the Discussion to emphasize this fact.

2) The authors concluded that Dlg regulates spindle orientation, but the images that they showed in most of their figures lacked some sort of Tubulin staining to show spindle orientation.

In some instances we used antibodies against centrosome components (centrosomin (cnn) or γ-tubulin) to measure spindle position. We have clarified this in the revised text.

3) Why didn't the over-expression of Dlg[S662D] rescue the spindle orientation defects as well as polarization of Miranda to the same extend as Dlg[wt] in Figure 3?

In neuroblasts expressing wild-type Dlg, the activity of Dlg is restricted to the apical cortex (basal Dlg is autoinhibited and apical Dlg is activated by aPKC). In neuroblasts expressing Dlg S662D, basal Dlg is active as well as apical Dlg. The ectopic Dlg activity leads to spindle misorientation. The revised manuscript contains a more thorough and careful explanation of this result.